A new genus of sinogaleaspids (Galeaspida, stem-Gnathostomata) from the Silurian Period in Jiangxi, China

Shan Xianren 1 2
Zhu Min 1 3 4
Zhao Wenjin 1 3 4
Pan Zhaohui 1
Wang Pingli 2
Gai Zhikun gaizhikun@ivpp.ac.cn 1 3 4
1 Key Laboratory of Vertebrate Evolution and Human Origins of Chinese Academy of Sciences, Institute of Vertebrate Paleontology and Paleoanthropology, Chinese Academy of Sciences , Beijing , China
2 College of Earth Science and Engineering, Shandong University of Science and Technology , Qingdao , China
3 CAS Center for Excellence in Life and Paleoenvironment , Beijing , China
4 University of Chinese Academy of Sciences , Beijing , China
Young Mark
Electronic publication date: 2020 May 15
Publication date: 2020
Volume: 8
Electronic Location ID: e9008
Received 2019 Jul 10; Accepted 2020 Mar 27
Copyright: ©2020 Shan et al.
Copyright year: 2020
Copyright holder: Shan et al.
License: This is an open access article distributed under the terms of the Creative Commons Attribution License, which permits unrestricted use, distribution, reproduction and adaptation in any medium and for any purpose provided that it is properly attributed. For attribution, the original author(s), title, publication source (PeerJ) and either DOI or URL of the article must be cited.
License URL: https://creativecommons.org/licenses/by/4.0/

Keywords: New genus, Galeaspida, Eugaleaspids, Phylogeny, Silurian

Funding: National Nature Science Foundation of China 41972006 41572108 41530102 Key Research Program of Frontier Sciences, CAS QYZDB-SSW-DQC040 National Program for support of Top-notch Young Professionals, and Strategic Priority Research Program of CAS XDB26000000 Innovation Training Programs for Undergraduates, CAS, and the State Key Laboratory of Palaeobiology and Stratigraphy (Nanjing Institute of Geology and Palaeontology, CAS, No. 193121) This work was supported by the National Nature Science Foundation of China (41972006, 41572108, 41530102), the Key Research Program of Frontier Sciences, CAS, Grant No. QYZDB-SSW-DQC040, the National Program for support of Top-notch Young Professionals, and Strategic Priority Research Program of CAS (XDB26000000), Innovation Training Programs for Undergraduates, CAS, and the State Key Laboratory of Palaeobiology and Stratigraphy (Nanjing Institute of Geology and Palaeontology, CAS, No. 193121). There was no additional external funding received for this study. The funders had no role in study design, data collection and analysis, decision to publish, or preparation of the manuscript.

==============================
Galeaspids are an endemic clade of jawless stem-gnathostomes known as ostracoderms. Their existence illuminates how specific characteristics developed in jawed vertebrates. Sinogaleaspids are of particular interest among the galeaspids but their monophyly is controversial because little is known about Sinogaleaspis xikengensis. Newly discovered sinogaleaspids from the Lower Silurian of Jiangxi, China provide a wealth of data and diagnostic features used to establish the new genus, Rumporostralis gen. nov., for Sinogaleaspis xikengensis. A morphological study showed that the sensory canal system of sinogaleaspids had mosaic features similar to those of three known galeaspids. There are 3–8 pairs of transverse canals in the Sinogaleaspidae, which suggests that the sensory canal system of galeaspid probably had a grid distribution with transverse canals arranged throughout the cephalic division. Phylogenetic analysis of Galeaspida supports the monophyly of the Sinogaleaspidae, consisting of Sinogaleaspis, Rumporostralis, and Anjiaspis. However, Shuyu and Meishanaspis form another monophyletic group, Shuyuidae fam. nov., which is outside all other eugaleaspidiforms. We propose a cladistically-based classification of Galeaspida based on our analysis.

Introduction

The Silurian–Devonian armored galeaspids were a prevalent and diverse clade of jawless stem-gnathostomes (Fig. 1A) that exhibited traits thought to belong to jawed gnathostomes (Gai et al., 2011; Gai & Zhu, 2012; Gai, Zhu & Donoghue, 2019). They contribute to our understanding of the conformation of the gnathostome body, which is significant to vertebrate evolution (Sansom, Rodygin & Donoghue, 2008). The family Sinogaleaspidae of the Eugaleaspidiformes within the Galeaspida, from the Lower Silurian of Xiushui, Jiangxi province and Changxing, Zhejiang province, is an important early clade (Figs. 1B–1D) possessing the characteristics that demonstrate the step-by-step transitions from jawless to jawed vertebrates. Synchrotron Radiation X-ray Tomographic Microscopy (SRXTM) provides an example of the cranial anatomy of Shuyu, a sinogaleaspid, and other important characteristics that may be compared with other early vertebrate groups (Gai et al., 2011; Gai & Zhu, 2012; Gai, Zhu & Donoghue, 2019). However, the phylogeny and morphology of its constituents are still disputed.

Figure 1 Phylogenetic placement (A) and interrelationships (B–D) of galeaspids.

(A) Galeaspids are attributed to the major armored, jawless fossil vertebrates (or ‘ostracoderms’, purple bar) (phylogenetic scenario adapted from Sansom et al. (2010), Artwork credit: Dinghua Yang and Nobu Tamura). (B–D) summary of previous hypotheses of galeaspid phylogeny showing controversy on the monophyly of Sinogaleaspidae (B simplified from Zhu, 1992; C simplified from Gai & Zhu, 2005; D simplified from Zhu & Gai, 2006; Gai et al., 2018).

The family Sinogaleaspidae was proposed by Pan & Wang (1980), and morphologically includes the species Sinogaleaspis, S. shankouensis, Meishanaspis and Anjiaspis, and ‘S.’ xikengensis, and ‘S.’ zhejiangensis (Pan & Wang, 1980; Pan, 1986a; Wang, 1991; Gai & Zhu, 2005; Gai, Zhu & Zhao, 2005; Liu et al., 2015). It remains unknown whether Sinogaleaspidae is a monophyletic group (Figs. 1B–1D) (Zhu, 1992; Gai, Zhu & Zhao, 2005; Zhu & Gai, 2006; Gai et al., 2018). Gai, Zhu & Zhao (2005) suggested that the three species assigned to Sinogaleaspis formed a paraphyletic group instead of a monophyletic group (Fig. 1C). S. shankouensis is more closely related to Yunnanogaleaspis and higher eugaleaspids than to ‘S.’ zhejiangensis and ‘S.’ xikengensis (Fig. 1C; Gai, Zhu & Zhao, 2005), whereas ‘S.’ zhejiangensis was determined as the sister to all other eugaleaspididiforms in later phylogenetic analyses (Fig. 1D; Zhu & Gai, 2006; Gai et al., 2018). Gai et al. (2011) determined the genus Shuyu for ‘S.’ zhejiangensis based on novel material, especially the three-dimensional images of the neurocrania. However, the systematic position of ‘S.’ xikengensis (Pan & Wang, 1980) is still unresolved due to its poor preservation and large amounts of missing data, especially related to its sensory canal system.

Five excavations of the Lower Silurian region of Xiushui, Jiangxi province have been organized since 2003. This location is the primary site of ‘S.’ xikengensis discovery and an abundance of Silurian fish remains have been found here, including sinogaleaspids, xiushuiaspids, and sclerites of dayongaspids and hanyangaspids. Our study describes newly discovered sinogaleaspid fossils and discusses their importance to the phylogeny of the Eugaleaspidiformes.

Geological Setting

The Silurian strata in the northwestern Jiangxi province are subdivided into six formations: the Lishuwo, Dianbei, Qingshui, Xiajiaqiao, Xikeng, and Xiaoxi formations (Zhao et al., 2009; Zhao & Zhu, 2010; Wang et al., 2018; Rong et al., 2019). New sinogaleaspid material was collected from two fossil sites in the Xikeng formation at Taiyangsheng Town, Xiushui County, Jiangxi province near Xikeng village (Fig. 2A) and a newly discovered location on the side of Shipan Reservoir (Fig. 2A). The Xikeng formation is mainly composed of medium- to thin-bedded yellow-green and purple siltstone and mudstone intercalated with fine sandstone. It is conformably underlaid by the Xiajiaqiao formation and unconformably overlaid by the Xiaoxi formation (Fig. 2B) (Wang et al., 2018). The galeaspids from the Xikeng formation include Sinogaleaspis shankouensis, ‘S’. xikengensis, Xiushuiaspis jiangxiensis, and X. ganbeiensis (Pan & Wang, 1980; Pan & Wang, 1983). The early vertebrate fossil assemblage was referred to as either the Sinogaleaspis–Xiushuiaspis assemblage (Pan, 1986b) or the Maoshan Assemblage (Zhao & Zhu, 2010) and it is consistent with the assemblage found in the Maoshan formation of the northwestern Zhejiang province (Pan, 1986b; Pan, 1988; Wang, 1991). The fish-bearing Xikeng formation is known as the Upper Red Beds (URBs) and is the equivalent of the Huixingshao formations in Chongqing and Guizhou, and the Maoshan formations in the Jiangsu and Zhejiang provinces. Although the precise age of the URBs in the western part of the Yangtze Platform is difficult to determine, it is thought to be from the middle-late Telychian period due to evidence from the underlying Xiushan formation with its invertebrate fauna and sequence stratigraphic analyses (Zhao et al., 2009; Zhao & Zhu, 2010; Wang et al., 2018; Rong et al., 2019). The age of the fish-bearing Xikeng formation is thought to be from the middle-late Telychian age of the Llandovery epoch during the Silurian period like those of the Huixingshao and Maoshan formations in South China (Fig. 2B).

Figure 2 Maps of the two fossil localities of Rumporostralis (A) and the fish-bearing lithological column (B) in Xiushui County, Jiangxi Province, China.

Material and Methods

The newly discovered sinogaleaspid material includes four head-shields of Rumporostralis xikengensis gen. nov. (IVPP V25136.1–4), and one head-shield of Rumporostralis shipanensis gen. et sp. nov. (IVPP V26114). All specimens are permanently housed in the collections of the Institute of Vertebrate Paleontology and Paleoanthropology (IVPP), Chinese Academy of Sciences and are accessible for examination. The holotypes of Sinogaleaspis shankouensis (GMC V1751) and Rumporostralis xikengensis (GMC V1753) are permanently housed in the collections of the Geological Museum of China (GMC) and were used for comparison and measurement.

All specimens were prepared mechanically using a Vibro-tool with a tungsten-carbide bit or a needle. Some specimens were reversed in latex casts. Specimens were measured with a digital vernier calliper, studied under optical zoom, and photographed with a Canon EOS 5D Mark III camera with a Canon macro photo lens (MP-E 65 mm 1:2.8 1–5×).

Results

Systematic paleontology

Subclass Galeaspida Tarlo, 1967

Order Eugaleaspiformes (Liu, 1965) Liu, 1980

Family Shuyuidae fam. nov.

Differential diagnosis. Shuyuidae differs from other families of Eugaleaspiformes in the splayed posterior supraorbital canals and absence of U-shaped median dorsal canals.

Type genus. Shuyu Gai et al., 2011

Referred genera. Meishanaspis

Remarks. Newly discovered sinogaleaspid material provides a wealth of data and reliable diagnostic features to erect the new genus Rumporostralis gen. nov. for ‘Sinogaleaspis’ xikengensis. Our phylogenetic analysis of Galeaspida shows that Shuyu and Meishanaspis are not included in Sinogaleaspidae and form another monophyletic group. A new family, Shuyuidae fam. nov., was created for Shuyu and Meishanaspis. Shuyuidae is positioned in the new cladogram as the sister to all other Eugaleaspididiformes with synapomorphies including a subtriangular head-shield and longitudinal oval or wedge-shaped median dorsal opening.

Family Sinogaleaspidae Pan & Wang, 1980

Differential diagnosis. Sinogaleaspidae differs from other families of Eugaleaspiformes in the V-shaped posterior supraorbital canals and more than 2 pairs of median transverse canals (autapomorphy).

Type genus. Sinogaleaspis. Pan & Wang, 1980

Referred genera. Rumporostralis gen. nov., Anjiaspis

Remarks. The amended Sinogaleaspidae, including Sinogaleaspis shankouensis, Rumporostralis xikengensis, (Figs. 3, 4) R. shipanensis (Fig. 5), and Anjiaspis reticularis is monophyletic with synapomorphy (U-shaped median dorsal canals) and autapomorphy (more than 2 pairs of median transverse canals).

Figure 3 Photographs of Rumporostralis xikengensis gen. nov.

A nearly complete external (A) and internal (B) mould of head-shield, holotype, GMC V1753A, B. (C) Close-up of coarse granular tubercles (180 degrees rotation of box region of Fig. 2A). (D) A nearly complete external mould of the head-shield, IVPP V25136.2a. (E) Close-up of the anterior part of head-shield (box region 1 of Fig. 3D). (F) Close-up of the posterior part of head-shield (box region 2 of Fig. 3D). (G) An incomplete internal mould of the head-shield, IVPP V25136.4. Abbreviations: br.c, branchial chamber; c, cornual process; ic, inner cornual process; md.o, median dorsal opening; nc.p, pore for the passage of the neural canal; orb, orbital opening; pi, pineal opening; pb.w, postbranchial wall; va.p, subcutaneous vascular plexus; vr, ventral rim.

Figure 4 Photographs (A) and interpretative drawing (B) of Rumporostralis xikengensis gen. nov., IVPP V25136.1 (C) close-up of postbranchial wall and pore on it for passage of the neural canal (box region of Fig. 4A).

Abbreviations: c, cornual process; ic, inner cornual process; ifc, infraorbital canal; ldc, lateral dorsal canal; ltc, lateral transverse canal; mdc, median dorsal canal; md.o, median dorsal opening; mtc, median transverse canal; nc.p, pore for passage of the neural canal; orb, orbital opening; pi, pineal opening; pb.w, postbranchial wall; soc1, anterior supraorbital canal; soc2, posterior supraorbital canal. ifc, infraorbital canal; ldc, lateral dorsal canal; ltc, lateral transverse canal; mdc, median dorsal canal.

Figure 5 Photograph and interpretative drawing of Rumporostralis shipanensis gen. et sp. nov.

(A) An incomplete internal mould of head-shield, holotype, IVPP V26114.1a, in dorsal view. (B) Interpretative drawing. (C) Close-up of the coarse granular tubercles (box region of Fig. 5A). Abbreviations: md.o, median dorsal opening; orb, orbital opening.

Genus Rumporostralis gen. nov.	

Etymology. Rumpo latin, state of being dehiscent or split; rostralis, Latin, snout, in referring to the rostral margin of the head-shield split by the anterior end of median dorsal opening.

Type species. Rumporostralis xikengensis (Pan & Wang, 1980)

Referred species. Rumporostralis shipanensis gen. et sp. nov.

Differential diagnosis. Rumporostralis differs from other Eugaleaspiformes by an unclosed rostral margin of the head-shield, indicating autapomorphy.

Remarks. The genus including R. xikengensis and R. shipanensis is uniquely characterized by the unclosed rostral margin.

Holotype. A nearly complete head-shield GMC V1753A, B

Referred specimens. A nearly complete head-shield IVPP V25136.1, three incomplete head-shields IVPP V25136.2–4.

Type locality and horizon. Xikeng village and Shipan reservoir, Taiyangsheng Town, Xiushui County, Jiangxi Province, China; Xikeng formation, Telychian, Llandovery, Silurian.

Differential diagnosis. R. xikengensis differs from the other species R. shipanensis in the small size of the head-shield.

Rumporostralis shipanensis gen. et sp. nov.	

Holotype. An incomplete head-shield IVPP V26114.1a, b

Type locality and horizon. Shipan reservoir, Taiyangsheng Town, Xiushui County, Jiangxi Province, China; Xikeng formation, Telychian, Llandovery, Silurian.

Differential diagnosis. R. shipanensis differs from the type species R. xikengensis by the large size of the head-shield.

Description

Rumporostralis xikengensis

Rumporostralis xikengensis is a small-sized sinogaleaspid with a subtriangular head-shield (Figs. 3, 4 and 6B). The rostral margin of the head-shield is disrupted by the anterior end of the median dorsal opening. The measurements of 4 specimens of R. xikengensis indicate that the size of the head-shield is consistent (Table 1). The head-shield is longer than it is wide with a length-to-width ratio of about 1:2. The head-shield protrudes caudally into a pair of cornual and inner cornual processes. The cornual processes are oriented caudo-laterally (or postero-laterally) and are short and rapidly taper off in the holotype and the newly discovered specimen IVPP V25136.1 (c, Figs. 3A, 3B, 4A and 4B). The inner cornual processes, which are completely preserved in the holotype and new specimen IVPP V25136.1 (ic, Figs. 3A, 3B, 4A and 4B), are small, spine-like, and caudally-oriented. The inner cornual processes are much smaller than the cornual processes.

Figure 6 Comparison of Sinogaleaspis shankouensis (A) and Rumporostralis xikengensis (B).

Artwork credit: Xiaocong Guo.

Table 1 Measurements of Rumporostralis xikengensis and R. shipanensis (mm).

Specimen	Rumporostralis xikengensis	R. shipanensis	
Item	V1753	V25136.1	V25136.2	V25136.3	V25136.4	V26114	
Maximum length of the head-shield	19.1	27.8	25.5	–	–	–	
Maximum width of the head-shield	17.8	21.5	19.1	16.6	–	63.0	
Length of the head-shield in midline	11.0	17.2	14.6	11.5	–	34.5	
Diameter of the orbital opening	1.7	2.5	1.8	1.7	2.1	5.8	
Distance between the orbital openings	3.3	4.4	4.3	2.8	4.0	8.0	
Long axis of the median dorsal opening	3.0	6.5	5.5	4.0	4.6	12.6	
Short axis of the median dorsal opening	1.1	1.3	1.1	1.0	1.4	4.9	
Length of the pre-pineal region in midline	5.5	10.6	6.8	6.5	9.2	18.5	
Length of the post-pineal region in midline	4.5	6.6	5.8	5.0	–	16.0	

The median dorsal opening (md.o) (Figs. 3, 4 and 6B) is fairly long and wedge-shaped or longitudinally elliptic in outline along the midline. The length-to-width ratio of the opening is less than 6 (Table 1). The anterior end of the median dorsal opening disrupts the rostral margin of the head-shield and its posterior end is positioned anterior to the level of the orbital opening (md.o) (Figs. 3, 4 and 6B).

The orbital openings are dorsally positioned on the head-shield (Figs. 3, 4 and 6B) and are round with a diameter of about 1.5 mm among the four specimens (Table 1). The orbital opening on the left side of specimen IVPP V25136.1 is longitudinal oval, which may be due to a deformation caused during preservation.

The pineal opening is clearly preserved in specimen IVPP V25136.2. It is level with the posterior margin of the orbital opening in the midline of the head-shield (Figs. 3D, 3E and 6B). The pineal opening is small and round with a diameter of 0.7 mm (Table 1). The ratio of the length of the pre-pineal and post-pineal region is about 1:2.

The sensory canal system is difficult to reconstruct in Rumporostralis xikengensis because it is preserved in only one specimen (IVPP V25136.1) (Figs. 4A–4B). The identified sensory canals consist of posterior supraorbital canals (soc2), infraorbital canals (ifc), lateral dorsal canals (ldc), lateral transverse canals (ltc), median dorsal canals (mdc), and median transverse canals (mtc) (Figs. 4A, 4B, 6B). The posterior supraorbital canals (Figs. 4A, 4B and 6B) are V-shaped. These canals originate from the anterior margin of the orbital opening, extend posteriorly along the inner side of the orbital opening, and meet behind the pineal opening. The median dorsal canals (Figs. 4A, 4B and 6B) are U-shaped and connect anteriorly with the posterior supraorbital canals level with the pineal opening and curve inward to converge with the opposite one on the midline of head-shield (Figs. 4A, 4B and 6B). The infraorbital canals (Figs. 4A, 4B and 6B) are an inverted S-shape. These canals originate on the lateral margin of the head-shield, pass through the lateral side of the orbital opening, and connect with the lateral dorsal canals. There are at least four pairs of lateral transverse canals (Figs. 4A, 4B and 6B) and three pairs of median transverse canals (Figs. 4A, 4B and 6B). The anterior three pairs of lateral transverse canals extend across the lateral dorsal canals to connect with the median transverse canals (Figs. 4A, 4B and 6B). The fourth lateral transverse canal is near the posterior edge of the head-shield and extends posterolaterally (Figs. 4A, 4B and 6B).

The endoskeletal roof of the oralobranchial chamber was poorly preserved in the internal mold of holotype GMC V1753B, but there are indications of at least 5 pairs of transversely elongated branchial fossae (Fig. 3B). Impressions for the subcutaneous vascular plexus are also preserved on the endoskeletal roof of the oralobranchial chamber in the internal mold of holotype GMC V1753B (Fig. 3B). There is an extensive endoskeletal postbranchial wall in specimen IVPP V25136.1, 2, (Figs. 3D, 4A and 4B) that closes the oralobranchial chamber posteriorly. The postbranchial wall is penetrated by a large pore in the midline of the head-shield for the passage of the neural canal to the body (Figs. 3D, 3F, 4A–4C).

The lateral margin of the head-shield is smooth and the surface of the head-shield is ornamented with closely set, coarse, granular tubercles (Fig. 3C). There are about 10 tubercles per square millimeter.

Rumporostralis shipanensis

Rumporostralis shipanensis is a medium-sized sinogaleaspid. The longest known head-shield is 52.4 mm; the widest known head-shield is 63.0 mm, and the length of its head-shield along the midline is 34.5 mm (Table 1). The rostral margin of the head-shield is unclosed. The holotype of this species is 12.6 mm along the long axis of the median dorsal opening (Figs. 5A and 5B) and 4.9 mm along the short axis. The diameter of the orbital opening (Figs. 5A and 5B) is 5.8 mm in the holotype. The orbital opening on the left side is a longitudinal oval, which may be due to a deformation during preservation. The distance between the paired orbital openings is 8.0 mm in the holotype. The lateral margin of the head-shield is smooth and the exoskeleton of the head-shield is ornamented with closely set, coarse granular tubercles (Fig. 5C). There are about 1.5 tubercles per square millimeter.

Comparison

Rumporostralis is similar to Shuyu, Meishanaspis, Anjiaspis, and Sinogaleaspis with a subtriangular head-shield, a wedge-shaped or longitudinal oval-shaped median dorsal opening, and spine-like cornual and inner cornual processes. It is comparable to Sinogaleaspis in its V-shaped posterior supraorbital canals, U-shaped medial dorsal canal, and at least 3 pairs of median transverse canals (Figs. 6A, 6B and 7). It is markedly different from Sinogaleaspis in the following aspects: (1) the head-shield of Rumporostralis is more slender than that of Sinogaleaspis, which has a wider-than-long head-shield (Figs. 6A, 6B and 7); (2) the posterior end of the median dorsal opening is in front of the level of orbital openings in Rumporostralis but behind the level of the center of orbital openings in Sinogaleaspis (Figs. 6A, 6B and 7); (3) the anterior end of the median dorsal opening reaches the anterior margin of the head-shield and disrupts the margin in Rumporostralis, but in Sinogaleaspis it separates from the rostral margin of the head-shield (Figs. 6A, 6B and 7); (4) the pineal opening is level with the posterior margin the orbital opening in Rumporostralis but the opening is more posterior in Sinogaleaspis (Figs. 6A, 6B and 7); (5) the ornamentations of Rumporostralis are coarse granular tubercles (Figs. 6B and 7) but are tiny granular tubercles in Sinogaleaspis (Figs. 6A and 7). There are some differences in the sensory canal systems of Rumporostralis and Sinogaleaspis (Figs. 6A, 6B and 7). For example, there are 3 pairs of median transverse canals identified in Rumporostralis xikengensis (Figs. 6B and 7) but up to 6 pairs have been identified in Sinogaleaspis shankouensis (Figs. 6A and 7) whose sensory canal system was comprehensively reconstructed based on 11 specimens (Gai et al., 2020). It is difficult to determine whether these differences are caused by the incomplete preservation of Rumporostralis or from another cause.

Figure 7 Life restoration of Sinogaleaspis shankouensis (left) and Rumporostralis xikengensis (right) in a fresh river.

Artwork credit: Xiaocong Guo.

Rumporostralis shipanensis resembles Rumporostralis, R. xikengensis in its unclosed rostral margin, the position of the median dorsal opening more anteriorly than the orbital openings, and its ornamentation of coarse granular tubercles. However, the head-shield of the former is about three times larger than the latter. Growth variations have been observed in some tessellate osteostracans, notably Escuminaspis and Superciliaspis (Sansom, 2007; Hawthorn, 2008; Keating, Sansom & Purnell, 2012; Scott & Wilson, 2012), but the relative size of the head-shield seems to be a reliable species diagnostic tool in osteostracans and galeaspids (Moy-Thomas & Miles, 1971; Janvier, 1996). Galeaspids have no cranial tesserae with ringed growth lines and have no clear growth series observed from the rich collection of Shuyu zhejiangensis in Zhejiang and Sinogaleaspis shankouensis in Jiangxi (Gai, Zhu & Zhao, 2005; Gai et al., 2020). There is only about a 10 percent variation in the length of the head-shield which indicates that the armored galeaspids either could not grow or became ossified only when they reached their definitive adult size, which is typical of most osteostracans (Westoll, 1945; Denison, 1952; Moy-Thomas & Miles, 1971; Janvier, 1996). The large size difference (up to 300 percent variation in head-shield length) between R. xikengensis and R. shipanensis is beyond the known range of intraspecific differences.

Phylogenetic analysis and results

We conducted a new phylogenetic analysis to explore the position of Rumporostralis and Sinogaleaspis within Galeaspida based on its known dataset (Gai & Zhu, 2005; Zhu & Gai, 2006; Gai et al., 2018) and new data from Sinogaleaspidae (Gai et al., 2020). Three taxa and five new characteristics were added to the data matrix presented by Gai et al. (2018) (File S1). The newly added characteristics are as follows:

[55] The unclosed rostral margin: (0) absent; (1) present.

[56] The ornamentation of the head-shield: (0) star-shaped tubercles; (1) tiny granular tubercles; (2) coarse granular tubercles.

[57] Pineal organ: (0) on front of or level with the posterior margin of orbital opening; (1) behind posterior margin of orbital opening.

[58] Pre-pineal region longer than the post-pineal region in mid-line of head-shield: (0) absent; (1) present.

[59] Preorbital commissure: (0) absent; (1) present.

Phylogenetic data entry and formatting were performed using Mesquite version 3.6 (Maddison & Maddison, 2015). The phylogenetic analysis was conducted using PAUP 4.0b with the parsimony analysis package and the heuristic search option (1000 replicates, random addition sequence) (Swofford, 2003). All characteristics were unordered and weighted equally, as in the earlier versions of this dataset. An early plesiomorphic osteostracan Ateleaspis was selected as the outgroup for our phylogenetic analysis (Sansom, 2009). Our analysis yielded 6 equivalent most-parsimonious trees (Fig. 8) with a tree length of 160, consistency index (CI) of 0.4562, and retention index (RI) of 0.7825 (File S2).

Discussion

Taxonomy of Eugaleaspidiformes

The Sinogaleaspidae family was established by Pan & Wang (1980), and five species Sinogaleaspis shankouensis, Rumporostralis xikengensis, Shuyu zhejiangensis, Meishanaspis lehmani, and Anjiaspis reticularis are morphologically assigned to this family (Pan & Wang, 1980; Pan, 1986a; Pan, 1986b; Wang, 1991; Zhu, 1992; Gai & Zhu, 2005; Liu et al., 2015). The previously morphologically recognized Sinogaleaspidae was paraphyletic in our cladogram. We restricted Sinogaleaspidae including Sinogaleaspis shankouensis, Anjiaspis reticularis, Rumporostralis xikengensis, and R. shipanensis (Fig. 8) and the amended Sinogaleaspidae are characterized by U-shaped median dorsal canals, V-shaped posterior supraorbital canals, and more than two median transverse canals. Sinogaleaspidae is more closely related to Yunnanogaleaspis and higher eugaleaspids by their typical eugaleaspid-pattern sensory canals. However, Shuyu zhejiangensis and Meishanaspis lehmani are not included in Sinogaleaspidae and form another monophyletic group altogether. We named the clade to a new family, Shuyuidae fam. nov. (Fig. 8). The new clade represents the deepest branching of the Eugaleaspidiformes with more primitive characteristics seen in the outgroup of Eugaleaspidiformes, such as the absence of the typical eugaleaspid-pattern sensory canals, splayed posterior supraorbital canals, and more lateral transverse canals from the infraorbital canals.

Figure 8 Strict consensus tree of 6 most parsimonious trees and cladistically-based classification of the Galeaspida.

Tree length = 160, consistency index (CI) = 0.4562, retention index (RI) = 0.7825, Numbers on branches denote bootstrap frequencies (above node) and Bremer support values (below node), bootstrap frequencies below 50 are not shown, analysis based on the dataset revised from (Zhu & Gai, 2006; Gai et al., 2018). Artwork credit: Dinghua Yang.

Eugaleaspidae was established by Liu (1980) to replace Galeaspidae for the junior homonym between the generic names Galeaspis in Agnatha and Trilobita (Liu, 1965). In the first cladistically-based classification of the Galeaspida, Zhu & Gai (2006) proposed that the species of Eugaleaspidae include Eugaleaspis changi (Liu, 1965), E. xujiachongensis (Liu, 1975), E. lianhuashanensis (Liu, 1986a), E. xiushanensis (Liu, 1983), Yunnanogaleaspis major (Pan & Wang, 1980), and Nochelaspis maeandrine (Zhu, 1992; Zhu & Gai, 2006). Zhu et al. (2012) described a new member of Eugaleaspidae Dunyu longiforus from the Ludlow (Silurian) Kuanti Formation of Qujing, Yunnan, and suggested that E. xiushanensis from the middle-late Telychian Huixingshao formation of Chongqing should be reassigned to Dunyu based on new data from the type specimen. Liu (1986a) and Liu (1986b) established Tridensaspidae for Tridensaspis magnoculus after considering its highly specialized rostral process and lateral projecting process which differed from other Eugaleaspidiformes. Our phylogenetic analysis indicated that Pterogonaspis yuhaii should be reassigned to Tridensaspidae to form a monophyletic group (Fig. 8) as it is more closely related to Tridensaspis magnoculus with the synapomorphies of the long rostral process and lateral projecting cornual process as in previous parsimony-based cladograms (Zhu, 1992; Gai, Zhu & Zhao, 2005; Zhu & Gai, 2006). Tridensaspidae is more closely related to Dunyu plus Eugaleaspis than Yunnanogaleaspis and Nochelaspis (Fig. 8). The continued grouping of Yunnanogaleaspis and Nochelaspis to Eugaleaspidae as proposed by Zhu & Gai (2006), will cause Eugaleaspidae to be a polyphyletic group and the family Eugaleaspidae should be further modified to represent a morphologically inclusive group. Therefore, we propose the removal of Yunnanogaleaspis and Nochelaspis from Eugaleaspidae (Fig. 8) to maintain the diagnostic stability of Eugaleaspidae which is mainly based on Eugaleaspis (Liu, 1965).

Sensory canal system of Sinogaleaspidae

The sensory canal system, also called the lateral line system in modern aquatic vertebrates, is a system of sense organs that serves to detect movements, vibration, and pressure gradients in the surrounding water (Bleckmann, 1993; Coombs & Montgomery, 1999; Bleckmann & Zelick, 2009; Mogdans & Bleckmann, 2012). It is unique to aquatic vertebrates from cyclostome fishes (lampreys and hagfish) (Fernholm, 1985) to amphibians (Schlosser, 2002). It is prevalent in the armored jawless fishes such as galeaspids, osteostracans, and heterostracans, and jawed placoderms during the Silurian-Devonian period (Fig. 9) (Jollie, 1962; Piveteau, 1964; Denison, 1978; Northcutt, 1985; Liu, 1986b; Northcutt, 1989; Janvier, 1996). The sensory canal system of galeaspids exhibits a characteristic festooned pattern consisting of two pairs of longitudinal stems and a varied number of transverse canals issuing from the stems (Liu, 1986b; Janvier, 1996). Its general pattern is comparable with other vertebrate groups. For example, most stem canals such as supraorbital canals, median dorsal canals, infraorbital canals, and lateral dorsal canals have their corresponding homologous parts in lampreys, heterostracans, osteostracans, and placoderms (Figs. 9A–9J) (Stensiö, 1932; Ritchie, 1967; Elliott, 1984; Northcutt, 1985; Liu, 1986b; Janvier, 1996). The number, placement, and branching pattern of the sensory canals in galeaspids varies significantly among different groups, even if the species are closely related (Figs. 9E–9J) (Liu, 1986b). Three patterns of sensory canals are generally recognized in galeaspids: (1) two median transverse canals with more lateral transverse canals issuing from the infraorbital canals and undeveloped supraorbital canals as in plesiomorphic taxa Dayongaspidae, Hanyangaspidae, and Xiushuiaspidae (Fig. 9E and 9G); (2) a V-shaped posterior supraorbital canal and one median transverse canal (dorsal commissures) as in Huananaspiformes and Polybranchiaspidiformes (Figs. 9H and 9J); (3) the U-shaped median dorsal canals anteriorly fused with the posterior supraorbital canals as in Eugaleaspidiformes (Figs. 9F and 9I) (Liu, 1986b; Wang, 1991).

Figure 9 The sensory canal system in early vertebrates.

(A) Heterostracan Anchipteraspis crenulata (redrawn from Elliott, 1984). (B) Petromyzontid Lampetra fluviatilis (redrawn from Stensiö, 1932). (C) Osteostracan Ateleaspis tessellate (redrawn from Ritchie, 1967). (D) Placoderm Radotina prima (redrawn from Denison, 1978). (E–J) Galeaspids: (E) Dayongaspis hunanensis (redraws from Pan & Zeng, 1985); (F) Sinogaleaspis shankouensis (redrawn from Gai et al., 2020); (G) Hanyangaspis guodingshanensis (redrawn from P’an, Wang & Liu, 1975); (H) Laxaspis qujingensis (redrawn from Liu, 1975); (I) Eugaleaspis changi (redrawn from Liu, 1965); (J) Sanchaspis magalarostrata (redrawn from Pan & Wang, 1981). Abbreviations: c, cornual process; cc, central canal; ic, inner cornual process; ifc, infraorbital canal; ldc, lateral dorsal canal; lf, lateral field; ltc, lateral transverse canal; mdc, median dorsal canal; md.o, median dorsal opening; mf, median field; mtc, median transverse canal; nhf, naso-hypophysial foramen; no, nasal opening; orb, orbital opening; pi, pineal opening; poc, preorbital commissure; soc, supraorbital canal; soc1, anterior supraorbital canal; soc2, posterior supraorbital canal; ro, rostral process; v.mdc, vestige of median dorsal canal.

The sensory canal patterns of sinogaleaspids are different from those of all other known galeaspids. The sensory canals of sinogaleaspids are a typical eugaleaspid-pattern with a U-shaped median dorsal canal (Figs. 6 and 9F) which is a dignostic characteristic of Eugaleaspidiformes (Fig. 9I). The U-shaped median dorsal canals were thought to be lost in Polybranchiaspidiformes and Huananaspidiformes, but their vestiges are sometimes visible as a pair of short canals crossing with the dorsal commissure in Polybranchiaspis, Damaspis, and Laxaspis (Fig. 9H) (Liu, 1986a; Liu, 1986b). Sinogaleaspids also exhibit the mosaic features of two other known patterns. For example, they have two additional lateral transverse canals issuing from the infraorbital canals (ltca,b) (Fig. 9F), which may be regarded as a plesiomorphic characteristic of galeaspids, since 3-4 lateral transverse canals are found on the infraorbital canal in the plesiomorphic taxa such as Dayongaspidae, Hanyangaspidae, and Xiushuiaspidae (ltca–c) (Figs. 9E and 9G). The number of lateral transverse canals tends to decrease in later evolution, but the vestiges of these canals can sometimes be observed on the infraorbital canals in Eugaleaspis changi (Fig. 9I) and Laxaspis qujingensis (Fig. 9H). Sinogaleaspids bear the typical V-shaped posterior supraorbital canal (Figs. 6 and 9F) which is a derived characteristic uniquely shared by Polybranchiaspidiformes and Huananaspidiformes (Figs. 9H and 9J). The preorbital commissure and central canal in sinogaleaspids are also found in some members of Huananaspidiformes and Polybranchiaspidiformes including Laxaspis and Sanchaspis (Figs. 9F, 9H and 9J).

The sinogaleaspid sensory canal system is notable for the presence of more than two pairs of median transverse canals (Pan & Wang, 1980; Gai & Zhu, 2005) which has been questioned by Wang, (1991) and Liu, Gai & Zhu (2014). However, newly discovered sinogaleaspids confirm their presence (Gai et al., 2020). Among three genera referred to sinogaleaspids, Sinogaleaspis has 6 pairs of median transverse canals (Fig. 6A) (Gai et al., 2020), Anjiaspis has 8 pairs (Gai & Zhu, 2005), and Rumporostralis has at least 3 (Fig. 6B) and more likely 6 pairs, like Sinogaleaspis. Among about 80 described galeaspid species, this feature occurs uniquely in sinogaleaspids but is very common in heterostracans (Fig. 9A; Kiaer, 1932; Kiaer & Heintz, 1935; Stensiö, 1964; Blieck, 1984; Liu, 1986b). Liu (1986b) observed a general grid-like pattern of the sensory canal system for plesiomorphic vertebrates composed of 2–3 pairs of longitudinal stems linked by transverse branches; this is a common pattern among the different types of sensory canal systems in various vertebrate groups. The sensory canal system of heterostracans is regarded as the ideal model for a general pattern (Blieck, 1984; Liu, 1986b).

The median transverse canals of sinogaleaspids occur in the post-orbital region of the head-shield and are level with the anterior, central, and posterior margins of the orbital opening as in Anjiaspis and Sinogaleaspis (Gai & Zhu, 2005; Gai et al., 2020). The grid distribution of the sensory canal system on the dorsal side of the head-shield in sinogaleaspids is made up of 4 longitudinal canals intercrossed with 3–8 pairs of transverse canals, reflecting the assumed plesiomorphic condition of vertebrates.

Conclusion

The newly discovered sinogaleaspids from the Lower Silurian in Jiangxi, China provides a wealth of new data and reliable diagnostic features to assign the new genus, Rumporostralis gen. nov., to ‘Sinogaleaspis’ xikengensis. This in-depth morphological study determined that the sensory canal system of sinogaleaspids exhibits the mosaic features of three known galeaspid patterns. The presence of 3-8 pairs of transverse canals in Sinogaleaspidae suggests that the sensory canal system of galeaspids probably displayed a grid distribution with transverse canals arranged throughout the cephalic division. An extended phylogenetic analysis of Galeaspida corroborates the monophyly of Sinogaleaspidae, which consists of Sinogaleaspis, Rumporostralis, and Anjiaspis. Shuyu and Meishanaspis were excluded from Sinogaleaspidae to form the monophyletic group Shuyuidae fam. nov., which is the sister of all other Eugaleaspididiformes. We propose a cladistically-based classification of the Galeaspida.

Supplemental Information

File S1 Data Matrix for phylogeny

Click here for additional data file.

File S2 Character description

Click here for additional data file.

Supplemental Information 1 Tree description

Click here for additional data file.

We are grateful to R-D Zhao, X-H Lin, Z-X Sun and B-C Sun for their help with the field work, X-C Guo for drawing Figs. 6 and 7, and D-H Yang and N Tamura for permission to use their fish logo in Figs. 1 and 8.

Additional Information and Declarations

Competing Interests

Author Contributions

Data Availability

New Species Registration

The authors declare there are no competing interests.

Xianren Shan performed the experiments, analyzed the data, prepared figures and/or tables, authored or reviewed drafts of the paper, and approved the final draft.

Min Zhu and Zhikun Gai conceived and designed the experiments, performed the experiments, analyzed the data, prepared figures and/or tables, authored or reviewed drafts of the paper, and approved the final draft.

Wenjin Zhao performed the experiments, prepared figures and/or tables, authored or reviewed drafts of the paper, and approved the final draft.

Zhaohui Pan performed the experiments, authored or reviewed drafts of the paper, and approved the final draft.

Pingli Wang performed the experiments, prepared figures and/or tables, and approved the final draft.

The following information was supplied regarding data availability:

The raw sinogaleaspid material including four head-shields of Rumporostralis xikengensis gen. nov. (IVPP V25136.1–4) and one head-shield of Rumporostralis shipanensis gen. et sp. nov. (IVPP V26114) are permanently housed and accessible for examination in the collections of the Institute of Vertebrate Paleontology and Paleoanthropology (IVPP), Chinese Academy of Sciences. The holotypes of Sinogaleaspis shankouensis (GMC V1751) and Rumporostralis xikengensis (GMC V1753) are permanently housed in the collections of the Geological Museum of China (GMC). The raw data matrix, character description and tree description are available in the Supplemental Files.

The following information was supplied regarding the registration of a newly described species:

Publication LSID: urn:lsid:zoobank.org:pub:1B734F17-7EBF-467E-A8B5-E33E8B9282E4.

Shuyuidae fam. nov. LSID: urn:lsid:zoobank.org:act:DF109F8C-EA60-4FD4-A0BB-7CC1DD8950BA.

Rumporostralis gen. nov.LSID: urn:lsid:zoobank.org:act:83C7CDB2-0013-4BBC-BF40-344CB2F2DE73.

Rumporostralis gen. nov. rumporostralis shipanensis gen. et sp. nov. LSID: urn:lsid:zoobank.org:act:0589DD8B-DA17-4B74-B699-B94BDF106B56.

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
