# Peer review of "A new genus of sinogaleaspids (Galeaspida, stem-Gnathostomata) from the Silurian Period in Jiangxi, China"

_PeerJ, doi:10.7717/peerj.9008_

## Round 0.1 · original submission · Major Revisions

Dear authors,

I would like to apologise for the lateness of this decision. But I have now received all reviewers comments. Given their input, I have accepted the decision of ‘major revisions’.

I look forward to receiving your revised manuscript.

·

Basic reporting

see general author notes

Experimental design

see general author notes

Validity of the findings

see general author notes

Additional comments

Shan et al describe some new sinogaleaspid fossil material identifying new taxa. These taxa are placed in taxonomic framework, subjected to a phylogenetic analysis, and discussed in the context of galeaspid evolution. I am happy to recommend publication of this manuscript in PeerJ following some minor revisions (listed below).
1. If Shuyuidae is a new family, it will also need a taxonomic diagnosis.
2. Some aspects of the taxonomy would benefit from highlighting the key synapomorphies that distinguish taxa. For example, what are the unambiguous synapomorphies of the revised Sinogaleaspididae? Can the authors confirm that developed median dorsal canals and more than 2 pairs of median transverse canals are not observed in any other galeaspids? A remarks section for Rumporostralis would help to highlight the unambiguous and unique syanpomorphy defining the genus.
3. The final discussion and conclusions section makes some interpretations beyond the available data. Given the phylogenetic analysis and its taxonomic scope, it is possible to infer or describe the evolution of sensory line patterns WITHIN galeaspids, but it is certainly not possible to identify or infer ancestral conditions for vertebrates without a much bigger analysis including a broader range of clades. I suggest removing this final observation - the manuscript happily stands alone with out it.
4. I don't quite understand the choice of the phylogenetic matrix. The authors use Gai and Zhu 2005, but why not use Zhu and Gai 2006? The later has a broader range of genera included and a more exhaustive character list.

More minor points:
L36 accumulative is not the correct word
L32 are not is
L44 delete the
L64 excursions is not the right word
L140 acanthous might not the best word choice here
L206 "as for the type species" but this IS the type species. Did you mean to refer to a previous diagnosis?
L256 The extension of the anterior end of the median dorsal opening terminal on the head shield- this character needs consideration from a taxonomic perspective. Could it be an artifact of preservation? Some of your specimens show incompleteness in this region
L307 be careful of the word 'cracked'. This implies some sort of destructive force, when I think the intention is to describe the position of the median opening as reaching the anterior margin of the headshield
L353-L359 This is a fascinating subject, but needs more thorough data and consideration before definitive conclusions are drawn
L406 have not has
L476. Abbreviations
Figure 7. This is a beautiful figure. Given the discussion of sensory line patterns, it might make sense to include that information in this phylogeny some how.

Reviewer 2 ·

Basic reporting

Overall the study is well described and executed. However, the text would benefit from the use of clearer English throughout, perhaps a re-edit and read through, this would clarify some meanings and descriptions and enhance the flow of the text in places.

Experimental design

The study aims to describe two new galeaspid species and place them within the phylogenetic context of the Eugaleaspidiformes. I believe the text could put more emphasis on our previous gaps in knowledge and how their contribution alleviates this. All other methods are well descried, with a clear systematic palaeontology section.

Validity of the findings

no comment

Additional comments

Overview
The paper introduces and describes two new species of galeaspids from the Lower Silurian of Jiangxi, China. The two new taxon are notable as their median dorsal openings are positioned at the very anterior end of the head shield, creating a break in the rostral end of the shield. The presence of 3-8 pairs of transverse commissures in the new taxon supports the presence of these in sinogaleaspids, which is not seen in other galeaspids taxa. The authors then incorporate the new taxa into a phylogeny of Eugaleaspidiformes erecting 5 new characters and determine the monophyly of the Sinigaleaspidae (to the exclusion of Shuyu zhejiangensis and Meishanaspis lehmani).
Overall the study is well described and executed. However, the text would benefit from the use of clearer English throughout, perhaps a re-edit and read through, this would clarify some meanings and descriptions and enhance the flow of the text in places.
The study aims to describe two new galeaspid species and place them within the phylogenetic context of the Eugaleaspidiformes. I believe the text could put more emphasis on our previous gaps in knowledge and how their contribution alleviates this. All other methods are well descried, with a clear systematic palaeontology section.
Below are general comments on sections, or specific lines
Abstract: The abstract could be improved by a clearer introduction to the animals being studied and place the new galeaspids taxa into their wider context.
Introduction: More of a general introduction would help non-specialists. Perhaps a phylogeny of early vertebrates with an introduction into their anatomy (especially galeaspids anatomy) would make your paper more accessible to non-specialists. I think the text would also benefit from an introduction to galeaspids phylogenetics, what has been done in the past? Where are the Eugaleaspidiformes placed within the Galeaspida? What are the previous contradictions in their phylogenetic relationships that your study may alleviate?
L43: “we use the synchrotron X-ray tomographic microscopy (SRXTM) to characterize the cranial 
anatomy of Shuyu zhejiangensis and provide the crucial fossil evidence for the 
 culmination of stepwise anatomical changes towards crown gnathostomes “ This sentence is slightly misleading, as it reads as if you are doing this in this paper and you do not state that Shuyu is within the Sinogaleaspididae and thus the Eugaleaspidiformes. I would suggest stating that SRXTM has been used to reveal the cranial anatomy of Shuyu, a Sinogaleaspididae, which in turn has provided valuable characters and anatomy within the galeaspids and for comparison with other early vertebrate groups ( Gai et al… etc ) – thus highlighting the groups importance.
L46: perhaps detail the characters that demonstrate the step-by-step transition from jawless to jawed vertebrates – this would emphasize why this group is important.
L49-63: this section is bit dense with lots of taxonomic names – would a diagram help with demonstrating what the taxa look like or the conflicting phylogenetic position of taxa in past studies? See Sansom (2009), Randle & Sansom (2017), Glinskiy (2019) – they provide conflicting previous topologies – this helps to show where the controversy is and thus increases the importance of your study.
L54-55: is this paraphylic grouping based upon phylogeny or taxonomy?
L74-75: “The new 
materials of sinogaleaspids were collected”, try “The new sinogaleaspid material was collected”
L101: material not materials
L119: you say all characters are unordered but in SI character list it says character 4 is ordered?
L119: please don’t use the term “basal” not useful when talking about taxa phylogenetically, perhaps use “deepest branching” or “considered sister taxon to….”, or “plesiomorphic”
L121: why are these galeaspids selected as outgroup taxa? I think an explanation/justification here would be good. Is it based on previous interpretations of anatomy, previous phylogenetic studies, stratigraphy?
L139: “Small to medium-sized galeaspids 
” –this isn’t that informative when you have given no introduction to what the size range is for galeaspids – perhaps put a measurement or size range for the headshields, or state the size.
L142: “orbital opening a little big, round” – this doesn’t make sense
L160: “basal” – try “Shuyuidae are positioned as the sister to all other Eugaleaspididiformes, and lack the typical eugaleaspid-type sensory canal pattern”
Line 165: sp. Lartin - latin
L174: try “its posterior margin ends before the level of the orbital opening anterior margin”
L175: “orbital opening fair big, round, dorsal positioned” – doesn’t really make sense
L177: “sensory system developed” – developed compared to what?
L181: sp “chamber” – “chambers”
L246 & 248 sp. in the holotype
L252: with the long axis
L253: axis of the median dorsal opening 

L307: is cracked the correct work here? Cracked would suggest it is broken and not an anatomical feature.
L313: are not is
L328: in the following aspects:
L329: more slender not “much slenderer”
L338: exists
L338-343: what are these similarities differences
L350: “the relative size of the head-shield seems to be a reliable 
species diagnostic in osteostracans and galeaspids” – should size be used as a diagnostic feature…. Have you compared the relative proportions of some of the features? If is scales completely and has no anatomical distinguishing features should it be considered a different species?
L358: remove “of”
L367: “diversified” - diverse
L399: why was the monophyly questioned? Many of the galeaspids papers referenced are in chinese so it may be a good idea to explain why this was questioned? Was it taxonomically, phylogenetically? Why?
L401-403: “In our cladogram, the monophly of the Sinogaleaspidae is robustly supported, if we remove Shuyu zhejiangensis and Meishanaspis lehmani from this group (Fig. 7)” what support measures are there to support this, I think it would add support to this statement if you include branch support statistics?
L458: “Within galeaspids, the number, placement and branching pattern of the sensory canals vary significantly in different groups, even if the species are closely related” – interesting point, which is true in other ostracoderm clades i.e. heterostracans. I wonder then how phylogenetically informative sensory canal pattern characters are? 

L508: “but is very common in heterostracans” – heterostracans commonly have 3-4 median transverse commissures. Why do you think the sinogaleaspids have up to 8?
L505 & L530: on line 505 you mention 3-8 pairs median transverse commissures and on L530 you state its 6-8 pairs - this needs clarifying
Phylogenetic characters
Characters: Overall the character list could do with explanations of the character codings – if you have used other studies characters and character codings you need to reference these with the original character numbers, This makes it clearer and more reproducible. If you’ve made any changes again state what these are and why (see below for further comments).
Figures
Fig 1: Colour combination – red and brown aren’t the best especially for colour blind peeps, also a fair bit of white space – could you remove the bounding box around China and squash it up a bit better. Key could be clearer – it’s hard to see what things are, distinguish from the different rock types.
Fig. 7. Usually the tree length and number of most parsimonious trees the strict consensus is constructed from is reported in the figure legend. It would also be good to see the branch support statistics on the tree (such as bremer decay and bootstrap values)
Comments on Character List
Appendix I
Characters used for the phylogentic analysis of the Eugaleaspiforms by Gai et al.(2005)
All characters except Character 4 are unordered – why is 4 ordered and all others not?
1. Shape of median dorsal opening: (0) round; (1) transverse oval (width>lengh); (2) longitudinal oval (length>width).
2. Longitudinal oval dorsal opening: (0) not slit-like (length/width<5); (1) slit-like (length/width>5) – why have you used this – why 5?
3. Anterior end of median dorsal opening: (0) subterminal; (1) terminal; (2) far from rostral margin of shield.
4. Posterior end of median dorsal opening: (0) in front of or level with anterior margin of orbital opening; (1) between the centre and anterior margin of orbital opening; (2) between the centre and posterior margin of orbital opening. (ordered)
5. First median transverse canals (mtc1): (0) present; (1) absent. Doesn’t change the meaning of the character but conventionally (0) is absent and (1) present
6. Third median transverse canals (mtc3): (0) present; (1) absent.
7. Posterior supraorbital canals (soc2): (0) present; (1) absent.
8. Ltca: (0) present; (1) absent.
9. Ltcb: (0) present; (1) absent.
10. Ltcc: (0) present; (1) absent.
11. Ltc4: (0) present; (1) absent.
12. Median dorsal canals: (0) not developed; (1) developed. What does developed and not developed mean?
13. Corner: (0) present; (1) absent. Corner of what?
14. Extending direction of corner: (0) projeccting laterally; (1) projecting backward.
15. Inner corner: (0) present; (1) absent.
16. Shape of inner corner: (0) broad leaf-shaped; (1) spine-shaped.
17. Ratio between preorbital length and post-orbital length in mid-line larger than 0.9: (0) no; (1) yes. Again why 0.9 –what’s the significance of 0.9?
18. Ratio between preorbital length and post-orbital length in mid-line larger than 1.1: (0) no; (1) yes. Same as previous comment
19. Pineal organ: (0) on front of or level with posterior margin of orbital opening; (1) behind posterior margin of orbital opening.
20. Ratio between pre-pineal length and post-pineal length in mid-line of cephalic shield larger than 1.0: (0) no; (1) yes.
21. Rostral process: (0) absent; (1) present. You have switched the codings around here
22. Shape of cephalic shield: (0) nearly triangular; (1) nearly semicircular; (2) nearly oval; (3) nearly trapezoid.
23. Edge of cephalic shield: (0) serrated; (1) smooth.
24. Anterior supraorbital canals: (0) exitence; (1) absence. Should this be present/ absent?
25. V-shaped Posterior supraorbital canals: (0) yes; (1) no.
26. The rostral margin is splitted by the median dorsal opening: (0) yes; (1) no. Sp. split
27. The ornamentation of the head-shield: (0) star-shaped tubercles; (1) tiny granular tubercles (2)coarse granular tubercles
28. the number of mtc: (0)one (1) two; (2) more than two; Again why just more than two – would be good with character explanations!

Reviewer 3 ·

Basic reporting

I carefully read “A new genus of sinogaleaspids (Galeaspida, stem-Gnathostomata) from the Silurian of Jiangxi, China”. In this manuscript, Gai and colleagues describe new galeaspid materials from the Lower Silurian Xikeng Formation in Jiangxi Province, China. On the basis of these materials, Gai and colleagues: a) assign the form previously known as Sinogaleaspis xikengensis to a new genus Rumporostralis; b) name an additional species of Rumporostralis, R. shipanensis; c) provide general discussion of the lateral line morphology and small size variations in galeaspids; and d) present a new phylogenetic analysis of eugaleaspidiforms.

I find the components a–c sound and well reasoned. In particular, the sensory canals in galeaspids are among the topics ripe for comparison and discussion in a widely accessible international journal. Although comparison is rather superficial, and although this comparison should be enhanced with citations to new papers and the broader literature on the topic (such as King et al. Palaeontology 61, 325–358, and general discussion of dermal plate homology in stem gnathostomes), I welcome this contribution.

I recommend the authors consider some presentation problems identified in text and figures, and address some issues with the phylogenetic analysis. The manuscript is written with some clarity overall, but suffers here and there for the lack of better terminologies to describe morphology or the refusal to use simpler, clearer language. Diagnosis needs to be differential against coeval galeaspids and closely related, con-familial taxa, and should indicate which, if any, is an autapomorphy. Figure 1 has many problems. I suggest the authors use clearer legends for the stratigraphic column, and remove international borders in dispute, which have no relevance whatsoever to the content of the paper. For particular comments, please refer to the following sections and the file attached with my suggestions.

Experimental design

The analytical design has some flaws that need to be addressed.
1) I understand that there is little to do about the high taxon:character ratio in galeaspids, but some characters are clearly duplicates and others are non-independent of each other.
2) Appendix says character 4 is ordered, but the main text states all characters were treated unordered.
3) Characters 17 and 18 give weight to the taxa with ratio lower than 0.9 (scoring “0” for both characters, even though character 18 is dependent on 17).
4) Character 26 overlaps with character 3.
5) Character 28 repeats characters 5 and 6.

The text argues that it “robustly support” monophyly of sinogaleaspidiids. Any measure for this? Decay index or bootstrap?

Validity of the findings

It is difficult to assess taxonomic validity without differential diagnosis. Again, diagnosis needs to be differential against coeval galeaspids and closely related, con-familial taxa, and should indicate which, if any, is an autapomorphy. If difficult to do this in text, consider presenting a table comparing these forms.

Additional comments

Gai and colleagues use “basal” a lot as descriptive term. This is not recommended. Shuyuiids are equally "basal" as non-shuyuiid eugaleaspidiforms because one can simply rotate the tree at the node. Perhaps the authors mean that shuyuiids are nested outside all other eugaleaspidiforms?

Similarly, Taxa do NOT form a clade BY a synapomorphy. Taxa DO form a clade WITH a synapomorphy.

Figure 1: Different rock types cannot be distinguished because patterns are minute and all similar from fine sandstone to quartz sandstone. Please amend and differentiate better.
What are the differences between the sold and broken horizontal lines? Unconformity?
What are the vertical lines? Gaps?
What do the colors represent?
Fish legend on the stratigraphic column seems to be aligned with a green layer.
Academic journals are not suitable in showing international boundaries that are in disputes (inset having a large blank showing the southern islands and Taiwan, with a line drawn around them, which are not necessary to the context of the paper). PeerJ should not and cannot endorse any stance on international politics by publishing such map. Show the entire region in the map, or only show the mainland China.

Annotated reviews are not available for download in order to protect the identity of reviewers who chose to remain anonymous.

---

## Round 0.2 · Minor Revisions

Dear authors,

The two reviewers have given different recommendations, so I have decided upon ‘minor revisions’. Please look very carefully at the comments from reviewer 3.

I look forward to receiving your revised manuscript.

Reviewer 2 ·

Basic reporting

The manuscript is much improved - regarding grammar and language

The amended figures are a lot clearer and add value to the manuscript

Experimental design

no comment

Validity of the findings

no comment

Additional comments

The authors have taken the majority of the reviewers suggestions on board. I have 2 minor notes to make about the figures:

Figure 2 - Silurian not Siluric in stratigraphic column

Figure 7 - Perhaps state that the bremer values are above branches, whereas bootstrap values are below?

Reviewer 3 ·

Basic reporting

I read the revised version of the manuscript "A new genus of sinogaleaspids (Galeaspida, stem-Gnathostomata) from the Silurian of Jiangxi, China" and reviewed the responses submitted by the authors. I note that the authors put some efforts into incorporating our suggestions. However, I found some of these modifications introduced new sources of confusion, and in the end, problems have multiplied.

Major concerns include:

The most serious error is in "differential diagnoses." The authors did not understand the difference between synapomorphy and autapomorphy, and include in diagnosis traits like "synapomorphy with other eugaleaspiforms" or "plesiomorphy". Symplesiomorphic character can be used in differential diagnosis, but it makes no sense to include "plesiomorphy" without specifying what this plesiomorphic trait excludes. In any case, differential diagnosis must state which trait excludes what taxa. Some of these confusions may be clarified if the authors list (in addition to or parallel with diagnosis) parsimoniously constrained syn-/autapomorphies.

Description is repetitive and redundant, because the authors repeat the same information for two species of Rumporostralis. I have difficult time figuring out what is shared and what is not between the species. This part of the paper is still quite confusing, but needs not be. Please set the text in comparative framework, and do not just list morphological traits as they come to you. You said something once, why say it again?

In Description, the authors provide maximum and minimum values of measurements. This may be precise but not accurate: the authors describe the known range of size as a systematic trait, but this is just a sample from a much larger population. Also, in this format, I have no idea if these measurements kept in proportions or not. Why not present ratio values? Do they vary linearly?

Rostral margin "split" or "pierced" by the medial dorsal opening. These words are as poor choices as "cracked." The opening reaches the rostral margin. The margin is disrupted, or it remains unclosed.

Shuyuids as the "deepest" branching galeaspids -- this is still misleading. Non-shuyuid eugaleaspiforms comprise a clade equally deep as shuyuids. They are nested outside (or form a sister group to) all other eugaleaspiforms. They form one of the earliest diverging, and deepest branching eugaleaspiform clades, but they are not "the" deepest branching or earliest diverging.

Figure 1 is still difficult to decode, even with the legend. The lithographic symbols are too small to see, and the patterns blend in when multiple layers are stacked together. Please use more intuitive design.

Figure 3D, 4A, B -- "Neural canal" is supposedly preserved in these specimens, but the canal (nc.p) is unclear in pictures and I am not sure if this is simply an embayment of the posterior margin of the headshield.

Experimental design

No comment.

Validity of the findings

I think the authors have made improvement on the phylogenetic analysis, and the current version is sufficient.

Additional comments

Description is poorly executed here and there. Many grammatical comments were made on the pdf file, so please refer to those. Some major comments include:

Description
Redundant and repetitive. Re-organize information content for conciseness.

Throughout: "Oralopharyngeal"
Why not "oropharyngeal"?

Throughout: "naturally" "natural"
I have no idea what this means.

L 269 "...rapidly tapered off..."
Poor word choice. Rapidly means time. In this case, some alternatives should be considered, such as "acutely."

L308: "...pass through..."
Poor word choice.

L461: "...to cover a larger morphospace..."
This is not about morphospace occupation, and it is not ecomorphological context. A certain cladistically defined lineage is found to embrace greater morphological variations than previously anticipated. So morphospace itself is not expanded; rather, its systematic compositions were modified.

For more specific comments, see the pdf attached.

Annotated reviews are not available for download in order to protect the identity of reviewers who chose to remain anonymous.

---

## Round 0.3 · Minor Revisions

Dear authors,

I only sent your manuscript to the reviewer who gave it ‘major revisions’ in the previous round of review. Their decision is ‘accept’ but they still highlighted some linguistic issues (although noting it is far better than the original submission).

I have decided upon ‘minor revisions’, as I want to give you the chance to have the manuscript checked one final time, as PeerJ does not provide a free linguistic check service. Note that once you re-submit your manuscript I will accept as it has passed peer-review.

Reviewer 3 ·

Basic reporting

No comment

Experimental design

No comment

Validity of the findings

No comment

Additional comments

The authors carefully considered all the comments and questions raised, and addressed them adequately in the manuscript. The text could benefit further from editor's help for a clearer language, but the paper already reads much better than the initial submitted version. I have no further request.

---

## Round 0.4 · accepted · Accept

Dear authors,

Thank you for your work on the language issues. I am happy to 'accept' your manuscript.

You will be contacted by production staff to take your through the proofing stages.

Thank you again for choosing PeerJ, and I hope you will use us again as your publishing venue.